# The Dual Role of TRADD in Liver Disease: From Cell Death Regulation to Inflammatory Microenvironment Remodeling

**DOI:** 10.3390/ijms26125860

**Published:** 2025-06-19

**Authors:** Xueling Wang, Qiwen Tan, Di Zhang, Huan Cao, Shenghe Deng, Yu Zhang

**Affiliations:** 1Center for Liver Transplantation, Union Hospital, Tongji Medical College, Huazhong University of Science and Technology, Wuhan 430022, China; wangxueling1218@163.com (X.W.);; 2Department of Infectious Disease, Union Hospital, Tongji Medical College, Huazhong University of Science and Technology, Wuhan 430022, China; 3Department of Hepatobiliary Surgery, Union Hospital, Tongji Medical College, Huazhong University of Science and Technology, Wuhan 430022, China

**Keywords:** TRADD, liver disease, apoptotic, inflammatory

## Abstract

The global burden of liver diseases continues to rise, encompassing diverse pathologies such as viral hepatitis, alcohol-associated liver disease (ALD), metabolic dysfunction-associated steatotic liver disease (MASLD), and hepatocellular carcinoma (HCC). In recent years, TNFR1-associated death domain protein (TRADD), a pivotal adaptor molecule in the TNF signaling pathway, has been found to play a dual regulatory role in the pathogenesis of liver diseases. Through its death domain, TRADD binds to TNFR1 and dynamically recruits downstream factors (e.g., TRAF2, RIPK1, FADD) to form Complex I or IIa, thereby activating pro-survival or pro-apoptotic signals that dictate hepatocyte fate and modulate the inflammatory microenvironment. This review systematically summarizes the molecular structure and functional networks of TRADD, along with its mechanistic roles in liver diseases: in HCC, TRADD expression correlates with tumor differentiation and is regulated by miRNA targeting; in ALD and MASLD, TRADD-mediated apoptosis is closely linked to fibrotic progression; and in acute liver injury, TRADD signaling is modulated by factors such as HO-1 to mitigate damage. Furthermore, TRADD inhibitors and antisense oligonucleotides demonstrate therapeutic potential. This review highlights the clinical translational value of TRADD as a diagnostic, therapeutic, and prognostic biomarker for liver diseases, providing a theoretical foundation for future precision medicine strategies.

## 1. Introduction

The liver is a multifunctional organ that plays a pivotal role in maintaining metabolic homeostasis, immune tolerance, and toxin clearance [1]. Liver diseases represent a major global public health challenge, accounting for over 2 million deaths annually—approximately 4% of total global mortality [2]. The spectrum of liver diseases is broad, encompassing viral hepatitis (e.g., hepatitis B and C), alcohol-related liver disease (ALD), metabolic dysfunction-associated steatotic liver disease (MASLD), cirrhosis, and hepatocellular carcinoma (HCC). Throughout these pathological stages, hepatocytes endure persistent injury and repair stress [2,3]. In recent years, widespread hepatitis B vaccination and the advent of direct-acting antiviral agents for hepatitis C have reduced the disease burden of viral hepatitis. However, the incidence of metabolic liver diseases, such as MASLD and ALD, has risen sharply. Projections indicate that liver-related deaths and end-stage liver disease cases attributable to these conditions will increase exponentially in the coming decades [4,5]. Consequently, there is an urgent need to enhance preventive measures, early diagnostic strategies, and innovative therapeutic research to address this global health crisis. Studies have shown that although these diseases arise from diverse etiologies, their pathological progression is driven by four interconnected core processes: cell death, chronic inflammation, fibrotic remodeling, and dysregulated tissue regeneration [6]. In chronic liver disease, hepatocytes undergo a pathological shift from classical apoptosis to necroptosis and pyroptosis—a transition often mediated by the tumor necrosis factor (TNF)-α signaling pathway [7,8]. TNF-α, a key proinflammatory cytokine, is upregulated in multiple hepatic pathologies. Upon binding to its receptor tumor necrosis factor receptor 1 (TNFR1), TNF-α triggers a signaling cascade that serves as a central axis linking extrinsic stress to hepatocyte fate determination [9,10]. TNFR1 activation elicits highly plastic signaling responses. Under certain conditions, TNFR1 engagement initiates nuclear factor κB (NF-κB) and mitogen-activated protein kinase (MAPK) signaling, promoting cell survival, repair, and proinflammatory cytokine expression [11,12,13]. In other contexts, TNFR1 activation instead induces caspase-8-mediated apoptosis [14,15]. This dichotomous signaling switch depends on the involvement and regulation of a central adaptor protein: TRADD (TNFR1-associated death domain protein).

TRADD is a crucial adaptor protein in the TNFR1 signaling transduction pathway. On the one hand, it binds to TNFR1 via its C-terminal domain (DD) and recruits downstream molecules such as tumor necrosis factor receptor associated factor 2 (TRAF2), receptor interacting protein kinase1 (RIPK1), cellular inhibitor of apoptosis proteins 1 and 2 (cIAP1/2) to form a plasma membrane bound complex (Complex I), activating the NF-κB and MAPK signaling pathways to promote cell survival and inflammatory responses [16,17,18]. On the other hand, under specific stress conditions—such as sustained ROS accumulation or impaired ubiquitination following TNF stimulation—TRADD dissociates from Complex I and forms cytoplasmic complex (Complex IIa) with Fas-associated protein with death domain (FADD), activating caspase-8 to induce apoptosis [19]. This dual regulatory mechanism enables TRADD to play a crucial role in the processes of cell death, inflammation, and fibrosis in liver diseases. In HCC, TRADD expression is associated with tumor differentiation, promoting tumor growth by activating pro-survival signals [20,21]. In ALD, TRADD-mediated apoptosis is closely linked to the progression of liver fibrosis [22]. In acute liver injury (ALI), activation of the TRADD signaling pathway exacerbates hepatocyte apoptosis [23]. Notably, in MASLD, TRADD inhibits hepatocyte apoptosis through activating the NF-κB pathway, exerting a protective effect on the liver [24]. Recently, TRADD has emerged as a promising therapeutic target. Small-molecule inhibitors such as ICCB-19 and apostatin-1 (Apt-1), which modulate TRADD activity, have demonstrated significant efficacy in inflammatory and tumor models [25,26,27].

In summary, TRADD is increasingly recognized as a key player in liver disease pathogenesis. Its central role in TNF signaling and its ability to bridge cell death and immune-inflammatory responses provide a robust theoretical foundation for understanding liver disease mechanisms and developing targeted therapies. This review systematically examines the structure and function of TRADD, as well as its critical contributions to liver disease progression, offering insights into its potential as a therapeutic target.

## 2. Molecular Structure of TRADD

TRADD is a relatively small protein composed of only 312 amino acids, yet it harbors multiple functional domains that confer remarkable versatility. Its structural architecture includes a highly structured C-terminal DD [16], a conserved central region, and a largely unstructured N-terminal segment containing several potential regulatory sites. The TRADD DD adopts a canonical six-helix bundle fold, which is critical for mediating homotypic and heterotypic DD interactions [28]. The DD superfamily is one of the largest known structural domain superfamilies, traditionally classified into four subfamilies: the classical DD, death effector domain, caspase recruitment domain, and pyrin domain [29,30]. However, the C-terminal domain of TRADD features a novel DD variant containing a β-hairpin motif [31], representing a distinct subfamily—termed TDD—that expands the diversity of the DD superfamily.

When interacting with TNFR1, TRADD undergoes a conformational transition from a closed state, where the N-terminal domain (NTD) and DD interact via sites like F18 and I72, to an open state [26]. In this open state, the DD engages in homotypic interaction with TNFR1-DD, exposing the binding interface, while the middle of the α3 helix in the NTD twists and the angle of the α6 helix is fine-tuned, thereby unveiling the TRAF2 binding site composed of hydrophobic core residues such as L17 and V19 [32,33]. Concurrently, key residues in the N-TRADD, including tyrosine 16 and histidine 65 located on β1 and β2 strands, form the hydrophobic region (Region I) of the binding interface with C-TRAF2. serine 67, situated in the loop between β2 and β3 strands, though contributing less to surface area, plays a critical role in maintaining local conformational stability, as the serine 67 mutation results in a 92-fold reduction in binding affinity. The binding interface dominated by hydrophobic interactions, with partial hydrophilic contributions from these residues, mediates the high-affinity interaction between N-TRADD and C-TRAF2, serving as the key molecular basis for TRAF2 recruitment in the TNFR1 signaling pathway [32,33].

Notably, TRADD function is not solely governed by domain interactions but is also finely regulated by post-translational modifications, particularly phosphorylation. Upon TNF-induced TNFR1 activation, TRADD is recruited alongside RIPK1, TRAF2, and TAK1 in proximity to the DD and Complex I, which mediates MAPK and NF-κB activation to promote cell survival [34]. This pathway can be further modulated by the phosphorylation of Jak-1, Jak2, STAT-3, STAT5, and IκB kinase (IKK) [35,36,37]. Consequently, TRADD phosphorylation plays a crucial role in regulating complex assembly, signal bias, and disease relevance. Moreover, Guan et al. demonstrated that phosphorylation of the SXXE/D motif within the death domain is involved in TNFR1-TRADD complex formation and subsequent NF-κB activation. Phosphorylated S215LKD and S296LAE motifs are also essential for TRADD-mediated recruitment of FADD and RIPK [38]. In summary, the translocation of TRADD from TNFR1-associated Complex I to FADD-containing Complex IIa is tightly regulated in vivo. This process is precisely controlled through modulation of TRADD’s conformational state and electrostatic microenvironment, which collectively reshape its interaction network and ultimately determine the specific signaling output.

## 3. Functional Network of TRADD

TRADD serves as a pivotal adaptor protein in TNF receptor signaling pathways, playing a central role in mediating inflammatory responses, programmed cell death (including apoptosis and necroptosis), and cellular stress [25,39,40]. Beyond its canonical function as a core regulator of TNFR1 signal transduction, TRADD also acts as a critical nexus connecting multiple death receptors (DRs) and downstream adaptors. Its functional network exhibits remarkable plasticity, characterized by dual signaling outputs—pro-survival versus pro-apoptotic signals—as well as non-canonical roles (Figure 1).

### 3.1. Dualistic Nature of TRADD-Mediated Signal Transduction

The classical TRADD signaling cascade initiates with its specific binding to the intracellular DD of TNFR1, leading to the formation of dynamic signaling complexes. Spatiotemporal and compositional differences define two distinct complexes: the membrane-associated Complex I and the cytosolic Complex IIa [41]. Complex I is the primary signaling hub assembled upon TNFR1 activation, comprising RIPK, TRAF2, and cIAPs [42]. Within this complex, ubiquitination of RIPK1 facilitates the recruitment and activation of downstream TAK1 and IKK complexes, driving NF-κB activation to promote inflammatory cytokine expression, cell survival, and tissue repair [43,44,45]. TRADD acts as a scaffold in this process—lacking intrinsic enzymatic activity but orchestrating interactions via its DD for TNFR1 binding and its N-terminal region for TRAF2 recruitment [46].

However, when Complex I stability is compromised (e.g., due to impaired RIPK1 ubiquitination, cIAP depletion, or excessive ROS accumulation), TRADD dissociates from RIPK1 and translocates to the cytosol. There, it nucleates Complex IIa with FADD and pro-caspase-8 to initiate apoptosis [19,45,47]. This “signal bifurcation” represents a critical regulatory checkpoint in TRADD’s functional network. Upon activation of caspase-8, it can directly cleave and activate downstream effector caspases to initiate apoptosis [48]. Additionally, caspase-8 cleaves Bid to generate truncated-Bid (t-Bid), which translocates to the mitochondria, promoting the oligomerization of pro-apoptotic proteins such as Bax and Bak [49,50,51]. This leads to increased mitochondrial outer membrane permeability, cytochrome C release, and subsequent recruitment and activation of caspase-9, ultimately activating effector caspases like caspase-3 to induce apoptosis [52,53]. TRADD-dependent Complex IIa assembly is a pivotal apoptotic trigger, and its inhibition markedly attenuates tissue damage and inflammatory infiltration [54].

When caspase-8 is inhibited (e.g., by inhibitors like Z-VAD), RIPK1 competes with FADD to dissociate from Complex I and instead binds to RIPK3 via their RIP homotypic interaction motifs, forming necrosome complex (Complex IIb). This triggers RIPK3 phosphorylation and oligomerization [55,56,57]. Furthermore, in RIPK1-knockdown L929 and HT-22 cells, TRADD can directly interact with RIPK3 to form a complex, promoting RIPK3 oligomerization and phosphorylation [40,58]. Phosphorylated RIPK3 further activates mixed lineage kinase domain-like protein (MLKL), causing its translocation to the plasma membrane and pore formation, ultimately resulting in membrane rupture and necroptosis [59,60].

### 3.2. Non-Canonical Roles of TRADD in Cross-Receptor Signaling

Emerging evidence highlights TRADD’s role as a signaling integrator beyond TNFR1, particularly in Toll-like receptors (TLRs) and TNF-related apoptosis-inducing ligand receptor (TRAILR) pathways. TLRs are pattern-recognition receptors on antigen-presenting cells that modulate local inflammation and immune tolerance [61,62]. TRADD engages with TLR3/4 pathways via TRIF interaction to activate NF-κB, expanding its influence in innate immunity [63,64]. TRAIL selectively induces apoptosis in cancer cells by binding DR4 (TRAILR1) and DR5 (TRAILR2) [65,66,67,68], sparing normal tissues. TRADD fine-tunes TRAIL-induced death by modulating c-FLIP degradation or RIPK1 incorporation into the Death-Inducing Signaling Complex (DISC) [69]. Intriguingly, TRADD expression correlates with TRAIL sensitivity in HepG2 tumor cells, suggesting its potential as a target to overcome TRAIL resistance [70]. Additionally, a high-glucose (HG) environment upregulates X-box binding protein 1 (XBP1) expression, enabling its binding to the TRADD promoter and subsequent elevation of TRADD levels. Increased TRADD promotes NIMA related kinase 7 (NEK7) activation, thereby facilitating nucleotide-binding oligomerization domain-like receptor protein 3 (NLRP3) inflammasome assembly. This process drives Caspase-1 cleavage and activation, ultimately leading to downstream target protein gasdermin D (GSDMD) cleavage and the induction of pyroptosis [61].

## 4. Research Advances on TRADD in Various Liver Diseases

Liver diseases represent a major global health burden with persistently high morbidity and mortality rates, characterized by complex pathological mechanisms involving multiple signaling pathways. As a key adaptor molecule in death receptor signaling, TRADD plays a pivotal role in the pathogenesis of various liver diseases—including HCC, ALD, MASLD, ALI, viral hepatitis, and liver fibrosis—by regulating apoptosis, inflammatory responses, and fibrotic progression (Table 1). Recent advances in understanding TRADD’s molecular functions and associated pathways have highlighted its potential as a therapeutic target.

### 4.1. TRADD and HCC

HCC ranks as the sixth most common cancer globally and the third leading cause of cancer-related deaths [83], accounting for 90% of primary liver malignancies. With poor prognosis and rising incidence projected over the next three decades [84,85], HCC is strongly associated with aflatoxin exposure, hepatitis B/C (HBV/HCV) infections, and metabolic disorders [86]. Due to its asymptomatic early stages and limited treatment options for advanced cases, identifying early biomarkers and novel therapeutic targets is critical.

Notably, TRADD expression correlates with tumor differentiation: a study of 39 HCC patients revealed significantly higher TRADD positivity in poorly differentiated HCC (52.4%) compared to moderately/well-differentiated cases (22.2%) [20]. Furthermore, the TRADD-targeting antisense oligonucleotide (ASO TRADD) significantly inhibits HepG2 cell proliferation by downregulating TRADD expression and consequently blocking the TNFR1-mediated NF-κB anti-apoptotic signaling pathway. More importantly, ASO TRADD demonstrates synergistic anti-tumor effects when combined with proteasome inhibitors (such as MG132 or ALLN), achieving remarkable inhibition rates of 80–93%. This enhanced efficacy stems from dual mechanisms: (1) proteasome inhibitors suppress NF-κB activity to decrease anti-apoptotic proteins (e.g., Bcl-2) while upregulating pro-apoptotic factors (e.g., p53, Bax), thereby activating the intrinsic apoptotic pathway; and (2) the combination therapy coordinately disrupts anti-apoptotic signaling networks while simultaneously activating both extrinsic and intrinsic apoptotic pathways, creating an amplified apoptotic cascade effect [71]. Natural compounds like dehydrocrenatidine also exhibit anti-HCC effects by modulating TRADD-mediated death receptor pathways to induce apoptosis [72]. MicroRNAs (miRNAs) critically regulate TRADD expression. For instance, miR-149* directly targets TRADD mRNA to inhibit NF-κB signaling, exerting anti-HCC effects. TRADD upregulation in miR-149* knockout mice correlates with increased HCC susceptibility [21]. Similarly, miR-199a-5p promotes hepatocyte proliferation and liver regeneration by suppressing the TNF-α/TNFR1/TRADD/CASPASE8/CASPASE3 axis, offering novel therapeutic insights [73].

### 4.2. TRADD and ALD

ALD remains a leading preventable cause of liver-related morbidity and mortality worldwide, encompassing steatosis, steatohepatitis, progressive fibrosis, cirrhosis, and HCC. Its severe manifestation, alcohol-associated hepatitis, may progress to acute-on-chronic liver failure [87,88]. Alcohol metabolism generates toxic byproducts that trigger inflammatory cascades involving cytokines, chemokines, and reactive oxygen species (ROS), culminating in tissue damage [89]. In ALD, TNF-α/TRADD interaction activates extrinsic apoptosis and ROS-mediated hepatocyte death, stimulating TGF-β and PDGF production to drive stellate cell-mediated collagen deposition and extracellular matrix (ECM) remodeling. miR-124-3p attenuates alcohol-induced hepatocyte injury and fibrosis by suppressing TRADD and other pro-apoptotic genes, positioning TRADD as a potential ALD therapeutic target [22]. In acute liver injury induced by alcoholism, alcohol significantly increases plasma transaminase activity and enhances hepatic sensitivity to lipopolysaccharide (LPS). This injury is mainly characterized by polymorphonuclear cell infiltration, formation of necrotic foci and microabscesses, and increased apoptosis in liver tissue. These pathological changes were found to be closely associated with elevated levels of mRNA expression of pro-apoptotic regulatory factor/junction proteins such as TRADD. The protective factor epidermal growth factor (EGF) effectively attenuates hepatocyte apoptosis by inhibiting the expression of pro-apoptotic genes such as TRADD [74].

### 4.3. TRADD and MASLD

Formerly termed non-alcoholic fatty liver disease (NAFLD), MASLD affects ~32.4% of the global population and has become the fastest-growing HCC etiology in the U.S. due to rising obesity rates [90]. Its inflammatory subtype, metabolic dysfunction-associated steatohepatitis (MASH), involves enhanced cell death and regeneration, driving liver injury, fibrosis, and HCC-prone cirrhosis [28]. Studies have demonstrated that the hypoxia-associated factor (HAF) directly promotes TRADD transcriptional activation by binding to HAF-binding sites (HBS, consensus sequence CCCCRRCCCC) in the TRADD gene regulatory region. HAF-mediated upregulation of TRADD expression recruits components such as RIPK1 to activate the NF-κB pathway, facilitating phosphorylation and nuclear translocation of p65/p50 subunits, thereby suppressing hepatocyte apoptosis. In HAF-deficient mice, hepatocyte-specific HAF knockout leads to downregulated TRADD expression accompanied by reduced NF-κB pathway activity, ultimately resulting in increased hepatocyte apoptosis, dysregulated lipid metabolism, and enhanced susceptibility to HCC. These findings indicate that TRADD-mediated NF-κB survival signaling plays a protective role in MASH-associated liver cancer [24].

### 4.4. TRADD and ALI

ALI is a clinical syndrome with multiple etiologies, characterized by acute inflammatory responses in the liver induced by drugs, viral infections, alcohol intoxication, ischemia–reperfusion, and other factors. The pathological features include hepatocyte degeneration, necrosis, and apoptosis. Hepatocyte death triggers subsequent inflammation, leading to excessive deposition of extracellular matrix proteins. Consequently, liver injury often progresses to hepatitis and liver fibrosis, serving as a critical initiating factor for cirrhosis and HCC [91]. Recent studies have demonstrated that in ischemia–reperfusion (I/R)-induced ALI, heme oxygenase-1 (HO-1) attenuates hepatocyte apoptosis by inhibiting the formation of the TRADD/FADD/caspase-8 complex, suggesting the clinical potential of targeting TRADD [23]. In drug-induced ALI, various compounds (e.g., oxymatrine, sodium fluoride, and copper) can cause liver damage by activating TRADD-dependent apoptotic pathways [75,76,77]. Additionally, the probiotic Lactobacillus plantarum C88 protects against aflatoxin B1-induced liver injury by downregulating TRADD and FADD expression, thereby suppressing hepatocyte apoptosis and inflammatory responses [78]. Fucoidan from Fucus vesiculosus alleviates concanavalin A (ConA)-induced ALI by inhibiting both intrinsic and extrinsic apoptosis mediated by the TNF-α/TRADD/TRAF2 and JAK2/STAT1 pathways [79]. These findings collectively indicate that inhibiting TRADD signaling may represent a promising strategy for mitigating ALI.

### 4.5. TRADD and Viral Hepatitis/Liver Fibrosis

Viral hepatitis is an inflammatory liver disease caused by various hepatitis viruses, including hepatitis A, B, C, D, and E [92]. Among them, chronic infections with HBV and HCV can lead to liver fibrosis and HCC [93], posing a significant global public health burden [94,95]. However, the precise mechanisms by which these viruses regulate the development of liver diseases remain unclear. Studies have shown that the HCV nonstructural protein 5A (NS5A) binds to TRADD, blocking its interaction with FADD and impairing TRADD-mediated NF-κB activation, which may contribute to HCV pathogenesis by inhibiting hepatocyte apoptosis [81]. Similarly, a significant correlation exists between HBV-encoded X protein (HBx) integration and TRADD expression levels, suggesting a potential role in HBV-induced hepatocarcinogenesis [80].

Liver fibrosis is a pathological repair response following chronic liver injury, manifested by excessive deposition of ECM proteins, which can ultimately progress to cirrhosis and HCC [96,97]. The activation of hepatic stellate cells (HSCs) plays a pivotal role in promoting ECM accumulation during fibrogenesis [98]. Notably, gallic acid (GA) has been shown to alleviate fibrosis by inducing necroptosis of activated HSCs via the TNF-α/TRADD/RIP3 pathway [82], offering a potential therapeutic strategy for liver fibrosis.

## 5. Therapeutic Potential of TRADD as a Drug Target

Currently, TRADD has not yet entered clinical trials as an independent therapeutic target, but its mechanistic roles in various diseases have been extensively studied and have indirectly influenced the development strategies of related drugs (Table 2). Consequently, the development of TRADD inhibitors and their applications in personalized medicine have become research hotspots.

In recent years, researchers have developed various small-molecule inhibitors targeting TRADD. ICCB-19 and Apt-1 are two known TRADD inhibitors that bind to the N-terminal TRAF2-binding domain of TRADD, thereby inhibiting TRADD-mediated apoptosis and inflammatory responses. ICCB-19 has been shown to restore cellular autophagy function by inhibiting TRADD activity and exhibits significant neuroprotective effects in mutant tau protein-induced proteopathy models [26]. Meanwhile, Apt-1 markedly improves cardiomyocyte pyroptosis and cardiac function in diabetic cardiomyopathy models [61] and can suppress TNF-α-induced apoptosis and necroptosis, offering novel therapeutic strategies for inflammatory diseases and cancer [39]. Other small-molecule compounds, such as triazoloquinoxaline derivatives, have also been found to inhibit TNFα-induced NF-κB and apoptotic signaling pathways by blocking the binding of TNFα receptors to TRADD and RIP1, thereby attenuating TNFα-mediated cellular responses [105]. Additionally, TRADD-targeted antisense oligonucleotides significantly inhibit the proliferation of HepG2 cells. More importantly, the combination of ASO-TRADD with proteasome inhibitors (e.g., MG132 or ALLN) synergistically enhances anti-tumor effects in HCC, achieving inhibition rates of 80–93% [71]. Furthermore, cystic fibrosis transmembrane conductance regulator (CFTR) reduces inflammatory responses by promoting lysosomal degradation of TRADD and inhibiting the NF-κB signaling pathway [100]. Manikandan et al. screened peptide mimetics (MMs03918858 and MMs03927281) targeting the TRADD-TRAF2 interaction interface, which show promise as potential modulators of cardiovascular diseases by inhibiting TRADD-TRAF2 association in atherosclerosis. These peptide mimetics are currently being commercially procured for further preclinical and clinical studies [99]. This discovery provides new insights for developing TRADD degradation-based inhibitors.

Studies have demonstrated that TRADD is widely expressed in various cell types, but its expression levels and functional states vary significantly across diseases and individuals, making it a potential biomarker for disease diagnosis and treatment. In acute myeloid leukemia (AML), low TRADD expression is associated with poor prognosis, suggesting its utility as a prognostic biomarker [101]. Moreover, high TRADD expression in glioblastoma (GBM) correlates with worse progression-free survival, indicating that TRADD may promote chemoresistance in tumor cells by activating the NF-κB pathway [102]. TRADD expression patterns in rectal cancer (RC) and kidney renal papillary cell carcinoma (KIRP) have also been used to construct prognostic models, demonstrating their potential for predicting patient survival rates [103,104]. Additionally, upregulated TRADD expression in diabetic retinopathy further supports its critical role in inflammation and apoptosis [106]. TRADD may also serve as a biomarker for early screening. In intrahepatic cholangiocarcinoma (CCA), high TRADD expression is associated with unfavorable clinicopathological features and could become an important diagnostic and therapeutic indicator [107]. Analyzing TRADD expression levels may help identify high-risk patient populations, enabling early intervention and treatment.

## 6. Summary and Future Perspectives

As a core adaptor protein in the TNFR superfamily signaling pathway [108], TRADD exhibits remarkable multifunctionality and complexity in various pathological processes of liver diseases. On one hand, it determines hepatocyte fate by mediating apoptosis, necroptosis, and autophagy [109,110]; on the other hand, it influences the activation, migration, and remodeling of the hepatic immune microenvironment by regulating inflammatory signaling pathways such as NF-κB and MAPK [111,112]. TRADD plays pivotal roles in HCC, ALD, MASLD, acute liver injury, chronic hepatitis, and liver fibrosis, demonstrating its multi-tiered regulatory functions. However, the dualistic nature and high interactivity of TRADD signaling pose challenges for its therapeutic targeting, particularly regarding selectivity. Although small-molecule inhibitors (e.g., ICCB-19 and Apt-1) have shown therapeutic potential by modulating TRADD activity [26], their selectivity and side effects require further optimization. Structure-based design of specific inhibitors or agonists (e.g., compounds targeting the N-terminal TRAF2-binding domain or C-terminal death domain) may offer new therapeutic strategies. Additionally, precise regulation of TRADD expression or function using gene-editing or RNA interference technologies represents a promising research direction.

In summary, TRADD serves not only as a critical bridge connecting cell death and inflammation but also as a key to unraveling the heterogeneity and clinical progression of liver diseases. Its dual functionalities in liver diseases highlight immense biological complexity and clinical potential. Future research should focus on elucidating its regulatory networks and molecular intervention pathways, systematically analyzing the dynamics and plasticity of TRADD-mediated signaling, to facilitate the translation of basic research into clinical applications and provide novel perspectives for precision medicine in liver diseases.

## Figures and Tables

**Figure 1 ijms-26-05860-f001:**
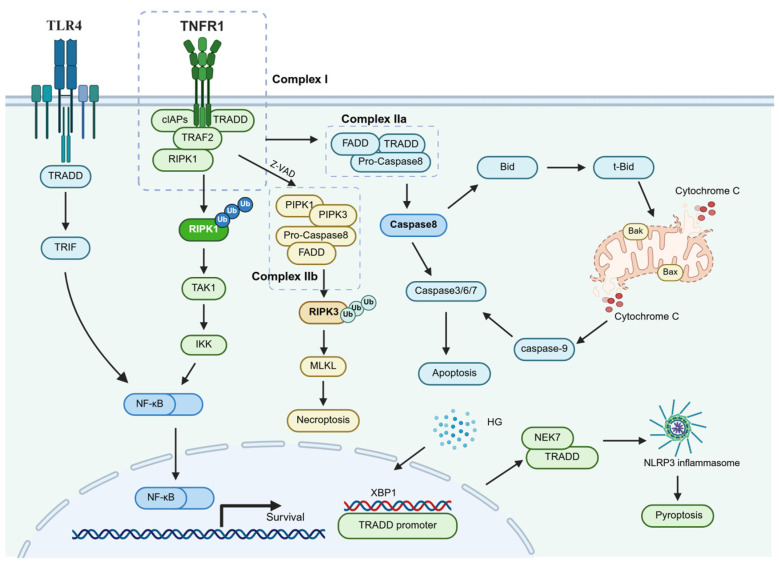
TRADD-mediated signaling pathways. TRADD acts as a central adaptor in TNF signaling, forming Complex I (pro-survival) or Complex IIa (apoptotic) to regulate cell fate. It also mediates necroptosis (Complex IIb) and pyroptosis, while integrating non-canonical pathways like TRAIL.

**Table 1 ijms-26-05860-t001:** The Role of TRADD in various liver diseases and related regulatory factors.

Liver Diseases	Role of TRADD	Related Regulatory Factors
HCC	TRADD expression correlates with tumor differentiation levels [20]. Targeting TRADD significantly inhibits HCC proliferation. Modulation of TRADD-mediated death receptor pathways suppresses HCC progression.	ASO-TRADD [71], miR-149* [21], dehydrocrenatidine [72], miR-199a-5p [73].
ALD	Inhibition of TRADD and other pro-apoptotic genes alleviates alcohol-induced liver injury and fibrosis.	miR-124-3p [22], EGF [74].
MASLD	Regulation of TRADD/RIPK1 transcription affects the NF-κB pathway.	HAF [24].
I/R Injury	Inhibition of TRADD-related complex formation reduces hepatocyte apoptosis.	HO-1 [23].
Drug-Induced Liver Injury	Activation of TRADD-dependent apoptotic pathways leads to liver damage.	Oxymatrine, sodium fluoride, and copper [75,76,77].
Aflatoxin B1-Induced Liver Injury	Downregulation of TRADD and FADD expression inhibits hepatocyte apoptosis and inflammation.	Probiotic Lactobacillus plantarum C88 [78].
ConA-Induced Liver Injury	Modulation of TRADD-mediated death receptor pathways alleviates liver injury.	Fucoidan [79].
Viral hepatitis	Significant correlation between HBx integration and TRADD expression levels. NS5A binds TRADD, block-ing its interaction with FADD and inhibiting NF-κB activation.	HBx [80], NS5A [81].
Liver Fibrosis	Induction of necroptosis in activated hepatic stellate cells (HSCs) via the TNF-α/TRADD/RIP3 pathway.	GA [82].

**Table 2 ijms-26-05860-t002:** Therapeutic strategies targeting TRADD and their applications.

Targeting Strategy	Mechanism	Associated Diseases
Small-Molecule Inhibitors	Binds to the N-terminal TRAF2-binding domain of TRADD, inhibiting apoptosis and inflammation. Restores cellular autophagy by inhibiting TRADD activity. Blocks TNFα receptor binding to TRADD, suppressing NF-κB signaling.	Neurodegenerative diseases [26], diabetic cardiomyopathy [61].
ASO-TRADD	Silences TRADD expression, inhibiting its pro-apoptotic effects. Synergistic effect with proteasome inhibitors (e.g., MG132 or ALLN) enhances antitumor activity.	HCC [71].
Peptide Mimetics	Disrupts TRADD-TRAF2 complex formation.	Cardiovascular diseases [99].
Bioactive Compounds	Promotes lysosomal degradation of TRADD, inhibiting NF-κB signaling.	Chronic lung infections [100].
Biomarker Potential	TRADD expression levels correlate with disease prognosis.	Various cancers (AML [101], GBM [102], RC [103], KIRP [104], CCA [100]).

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
