# Peer review of "The Dual Role of TRADD in Liver Disease: From Cell Death Regulation to Inflammatory Microenvironment Remodeling"

_ijms, 2025, doi:10.3390/ijms26125860_

Round 1

Reviewer 1 Report

Comments and Suggestions for Authors

This is an excellent review article that effectively summarizes the current understanding of the role of TRADD in liver disease. I have only a few suggestions to further improve the quality and completeness of the manuscript:

1. Please consider including a figure illustrating the emerging mitochondrial functions of TRADD discussed in Section 3.3. A visual summary would greatly enhance the reader’s comprehension of these recent insights.

2. While the review comprehensively covers necrosis and apoptosis, it would benefit from a brief discussion on the potential relevance of TRADD in other regulated cell death pathways, such as pyroptosis and ferroptosis.

Reviewer 2 Report

Comments and Suggestions for Authors

After a careful review of this document, I would like to propose the following main revisions to enhance its clarity, specificity, and overall quality:

1. Introduction: “ammatory cytokine, is up-53 regulated .......... This dichotomous signaling switch depends on the involvement and regulation of a central adap tor protein: TRADD (TNFR1-associated death domain protein).” The dual role of TRDDA is poorly articulated, and the differences in its roles at different pathological stages are not clarified.

2. Molecular Structure of TRADD section: “The interaction between TRADD and TNFR1 relies on key residues, including tyro-96 sine 16, histidine 65 The interaction between TRADD and TNFR1 relies on key residues, including tyro-96 sine 16, histidine 65, and serine 67.” This is just a list with no specific explanation of the binding mode and function.

3. 4.1 TRADD and HCC section: “Synergistic effects emerge when ASO-TRADD combines with proteasome inhib-201 itors (e.g., MG132 or ALLN), achieving 80 -93% tumor growth inhibition.” Suggest adding specific synergistic mechanisms.

4. 4.3 TRADD and MASL section: “Hypoxia-associated factor (HAF) modulates TRADD and RIPK1 transcription to regulate NF-κB activity, influencing MASH-to-HCC progression.” It is not clear how HAF binds to TRADD. Suggested addition.

5. Icons and citations: In Figure 1, the full names of “Complex I” and “Complex IIa” were not labeled when they first appeared, so it is suggested to add them; and the role nodes of TRADD in the mitochondrial pathway were not labeled, so it did not highlight TRADD; the references cited in the article are not current enough, so it is suggested to update and add them; the article cited references are not current enough. The references cited in the article are not up-to-date enough, so we suggest updating and supplementing them.

These suggestions are designed to improve the quality and readability of the article, while maintaining the rigor and rationality of the content. I hope these suggestions will be helpful to your research.
